# Speed Control of PMSM Based on Fuzzy Active Disturbance Rejection Control under Small Disturbances

## Qi Zhang * and Caiyue Zhang

Guilin University of Electronic Science and Technology, Guilin 541000, China; populusalba2020@outlook.com
* Correspondence: qi.zhang@guet.edu.cn

**Abstract:** Permanent Magnet Synchronous Motors (PMSMs), with their simple design, small size, and high-power factor, are ideally suited for realizing high-power AC drives and are widely used in various industries. In this study, Fuzzy Active Disturbance Rejection Control (Fuzzy-ADRC) is used to control the speed of the PMSM. When a slight external disturbance occurs, this control strategy maintains the suppression characteristics of the self-excited control for the disturbance and enhances its ability to compensate for the disturbance. First, a mathematical model was developed to study the surface mount PMSM. Then, a motor control simulation model was created using PI control, vector control, and other control methods. The verification results indicate that the improved Fuzzy-ADRC system performs well under both internal and external minor disturbances. It exhibits a faster dynamic response and reduced regulation time (0.026 s to 0.017 s) compared to the traditional ADRC system. Furthermore, it shows less overshoot (reduced from 70% to 2.9%) compared to the Sliding Mode Observer (SMO). Taken together, the improved Fuzzy-ADRC system is characterized by small steady-state error, high load capacity, and control accuracy. With the assistance of this control strategy, the system can track speed with high accuracy and possesses stronger anti-interference capability to mitigate load disturbances.

**Keywords:** active disturbance rejection control; fuzzy control; permanent magnet synchronous motor; speed control





## 1. Introduction

The PMSM is becoming increasingly widespread in industrial manufacturing due to its many advantages over other types of motors [1]. Many high-precision and high-performance control methods for the PMSM have been widely developed and applied, including PI control [2,3], adaptive control [4,5], Sliding Mode Control (SMC) [6,7], $H_\infty$ control [8,9], model-referenced adaptation [10,11], and Internal Model Control (IMC) [12,13]. Despite the widespread usage of PI control, it is challenging to obtain an optimal performance because PI controllers are highly dependent on parameters that are challenging to adjust in a disruptive and changing work environment. Adaptive control is governed by complicated laws. Conventional SMOs introduce high-frequency jitter due to the reference of discontinuous switching functions, and the use of a low-pass filter to filter out the high-frequency jitter means the estimated position is always delayed from the actual position. The application of these control techniques is constrained as a result of these issues.

In recent years, ADRC proposed by Prof. Kyung-Ching Han has been applied in the PMSM speed servo system [14–16]. It mainly consists of a Tracking Differentiator (TD), an Extended State Observer (ESO), and the Nonlinear State Error Feedback (NLSEF) control law [17]. The core of ADRC is the ESO. The ESO employs two parameter values to estimate the rotor speed and disturbance states of the system, with particular emphasis on estimating the total perturbation. This estimation ultimately impacts the control effectiveness. NLSEF, as a component of the ADRC algorithm, demonstrates superior response efficiency when compared to the linear control rate. The primary purpose of TD is to facilitate a transition

process that minimizes overshooting [14]. Furthermore, ADRC has been enhanced through the application of various intelligent control algorithms. In [18], an improved ADRC with ESO and finite-time stable TD was applied to the flexion–extension motion of the shoulder joint and the elbow joint. The method focuses on tracking accurate motion trajectories and reducing trajectory tracking errors to improve the tracking accuracy of the system and enhance the robustness of the controller to internal parameter uncertainties. In a previous study [19], a measurement delay-compensated linear ADRC method is proposed. This linear ADRC-based method addresses parameter perturbations, dead time nonlinearities, and unmodeled uncertainties to ensure system immunity performance. A previous study [20] introduced a hybrid control algorithm that combines Backpropagation Neural Network (BPNN)-based ADRC with adaptive fuzzy SMC. This approach combines fuzzy SMC with BPNN-based ADRC, eliminating the tuning blind zone and enhancing the system's anti-interference capacity, while also ensuring system adaptability. A parameter tuning method for ADRC using a Genetic Algorithm (GA) was proposed in reference [21]. According to experimental results, GA can handle difficult problems that are beyond the capabilities of conventional empirical techniques and demonstrates superior performance in adjusting the parameters of stabilized systems. In these improved ADRC algorithms, a single ESO is assigned the task of estimating and compensating for the total disturbance, leading to an increased burden on the ESO. To ensure precise estimation, the ESO must choose a substantial gain, which can potentially trigger system instability and vibration [22]. In [23], a load-adaptive dual-loop drive system for a robotic arm joint is described. This system utilizes an improved ADRC that integrates the position and velocity of the motor rotor and introduces a Fuzzy Controller (FC) for parameter tuning. This method can alleviate the burden of ESO prediction and adjustment to some extent. In a particular study [24], a dynamic parameter tuning method for ADRC is described, which involves the combination of GA and BPNN. GA is used to optimize the initial value of the BPNN to achieve real-time tuning of ADRC parameters. This method significantly improves the system control performance but requires the use of more complex algorithms and computational processes.

The traditional ADRC makes it difficult to achieve the required control effect for small perturbations and can only accurately predict and compensate for external perturbations with large fluctuations. To enhance the disturbance compensation capability of the conventional ADRC and achieve accurate motor speed tracking, this paper presents a Fuzzy-ADRC-based speed controller tailored for small disturbances. This approach is derived from the mentioned literature review. The main advantages of the improved ADRC are (1) enhanced robustness to a variety of external disturbances and internal uncertainty and a faster dynamic response. (2) An FC is added to the rotational speed feedback loop to proportionally amplify the speed error by introducing proportional coefficients. The amplified speed error, after passing through the ESO transformation, obtains a larger speed compensation. This enhances the compensation capability of ADRC under small disturbances. This strategy achieves the adaptive adjustment of disturbance compensation to ensure high-precision motor speed tracking and effectively enhances the robustness of the system.

In the second part of this paper, the SMO system is modeled, designed, and described. The third part discusses the design of a speed controller that incorporates ADRC and Feedforward Control (FFC) to enhance disturbance compensation. The fourth part showcases the experimental outcomes of simulating the Fuzzy-ADRC-based speed controller. These outcomes provide evidence of the efficacy of the described control strategy. Finally, the fifth part summarizes the entire paper and presents the research outlook.

Table 1 presents some of the terms covered in the article, including their corresponding abbreviations and full names.

**Table 1.** Summary of terminology and abbreviations.

| Technical Term | Acronyms |
|---|---|
| Internal Model Control | IMC |
| Sliding Mode Control | SMC |
| Genetic Algorithm | GA |
| Backpropagation Neural Network | BPNN |
| Fuzzy Controller | FC |
| Sliding Mode Observer | SMO |
| Feedforward Control | FFC |

## 2. SMO-Based PMSM Simulation Modeling

### 2.1. Motor Modeling

This research paper examines a PMSM with equal inductance in both the d-axis and q-axis of the motor rotor. Before establishing the mathematical model, the following assumptions are made: The magnetic saturation of the motor core is neglected, eddy currents and hysteresis losses are not counted, and the current in the motor is a symmetrical three-phase sinusoidal current [25]. The mathematical model for the motion and current equations of the PMSM can be derived using Equation (1) after Clarke and Park transformations:

$$
\begin{cases}
\frac{d\omega_m}{dt} = -\frac{T_L}{J} - \frac{B}{J}\omega_m + \frac{K_t}{J}i_q \\
\frac{di_d}{dt} = -\frac{R_s}{L_s}i_d + n_p\omega_m i_q + \frac{u_d}{L_s} \\
\frac{di_q}{dt} = -\frac{R_s}{L_s}i_q - n_p\omega_m(i_d + \frac{\varphi_f}{L_s}) + \frac{u_q}{L_S}
\end{cases}
\tag{1}
$$

where $i_d, i_q$ and $u_d, u_q$ represent the current and voltage of the PMSM rotor in the rotating coordinate system, respectively; $L_s, R_s$ is the inductance and resistance of motor stator; $\omega_m, n_p$ is the mechanical angular velocity and pole pair number; $J, \varphi_f$ is the rotational inertia and the flux linkage parameter; $K_t, B$ is the torque constant and friction coefficient; and $T_L$ is the load torque.

From Equation (1), it can be seen that $i_d, i_q$ and $\omega_m$ in the PMSM have a strong coupling relationship, and it is necessary to decouple $i_d$ and $i_q$ to obtain a high-precision control system. Since this paper focuses on the PMSM zero-low speed regulation problem under $i_d = 0$ control mode, there is no need to consider the coupling problem between $i_d$ and $i_q$. At this time, Equation (1) is converted to the following equation:

$$
\begin{cases}
\frac{d\omega_m}{dt} = -\frac{T_L}{J} - \frac{B}{J}\omega_m + \frac{K_t}{J}i_q \\
\frac{di_q}{dt} = -\frac{R_S}{L_S}i_q - \frac{n_p\varphi_f}{L_s}\omega_m + \frac{u_q}{L_s}
\end{cases}
\tag{2}
$$

In addition, the electromagnetic torque equation for PMSM is given in the following equation:

$$
T_e = K_t i_q = 1.5 n_p \varphi_f i_q
\tag{3}
$$

### 2.2. SMO Establishment

The SMO is a simple and effective control method for nonlinear systems with disturbances. It offers benefits such as excellent transient performance, a quick dynamic response, insensitivity to changes in system parameters and external disturbances, and robustness. In this section, the SMO is introduced to address the issue of the PI controller's dependence on system parameters, as well as the difficulty of parameter debugging. The PMSM control system relies on the disparity between the feedback speed and the set speed for its operation. Figure 1 is the overall schematic diagram of the SMO control system.

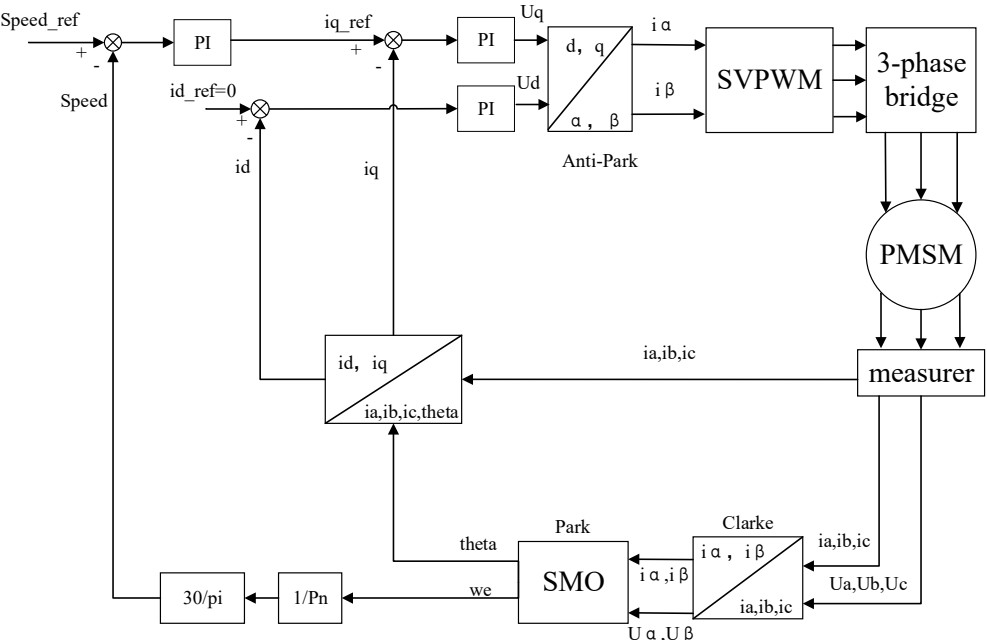

**Figure 1.** Block diagram of motor control structure based on SMO.

After the Clarke transformation, the motor's equations related to the $\alpha$ and $\beta$ axes can be expressed as follows:

$$\begin{bmatrix} u_\alpha \\ u_\beta \end{bmatrix} = \begin{bmatrix} R + pL_d & \omega_e(L_d - L_q) \\ -\omega_e(L_d - L_q) & R + pL_q \end{bmatrix} \begin{bmatrix} i_\alpha \\ i_\beta \end{bmatrix} + \begin{bmatrix} e_\alpha \\ e_\beta \end{bmatrix} \tag{4}$$

where $L_d$, $L_q$ is the inductance of the stator in the PMSM; $\omega_e$ is the electrical angular velocity; $i_\alpha$, $i_\beta$ and $u_\alpha$, $u_\beta$ are the current and voltage of the motor stator under $\alpha$-$\beta$, respectively; and $e_\alpha$, $e_\beta$ is the extended counter electromotive force (EMF), which meets:

$$\begin{bmatrix} e_\alpha \\ e_\beta \end{bmatrix} = \left[ (L_d - L_q)\left( \omega_e i_d - \frac{di_q}{dt} \right) + \omega_e \varphi_f \right] \begin{bmatrix} -\sin \theta_e \\ \cos \theta_e \end{bmatrix} \tag{5}$$

where $\theta_e$ is the electrical angle.

*2.3. Setting Up the PI Controller with Parameters*

To facilitate parameter tuning, the motion equation of the PMSM can be reformulated as follows:

$$J\frac{d\omega_m}{dt} = T_e - T_L - \xi\omega_m \tag{6}$$

where $\omega_m$ denotes the motor's mechanical angular velocity; $J$ is the rotational inertia; $\xi$ is the damping coefficient; and $T_L$ is the load torque.

According to the "active damping", the parameters of the speed loop PI controller are designed, and active damping is defined as:

$$i_q = i'_q - \xi_a\omega_m \tag{7}$$

Assuming the motor starts without any load, we can derive Equations (6) and (7) and combine them with Equation (3) due to the implementation of the control method $i_d = 0$:

$$\frac{d\omega_m}{dt} = \frac{1.5p_n\varphi_f}{J}i'_q - \frac{1.5p_n\varphi_f}{J}\xi_a\omega_m - \frac{\xi}{J}\omega_m \tag{8}$$

Shifting the poles of Equation (6) is necessary in order to achieve the desired closed-loop bandwidth:

$$\omega_m(s) = \frac{1.5 p_n \varphi_f}{J(s + \beta)} i'_q(s) \tag{9}$$

By applying the Rasch transform to Equation (8) and contrasting the results with Equation (9), one can determine the active damping coefficient $\xi_a$:

$$\xi_a = \frac{J\beta - \xi}{1.5 p_n \varphi_f} \tag{10}$$

The velocity loop controller in this paper is a conventional PI regulator, and its expression is given as follows:

$$i_q{}^* = (K_{pw} + \frac{K_{iw}}{s})(\omega_m{}^* - \omega_m) - \xi_a \omega_m \tag{11}$$

So far, the relevant control parameter $K_{pw}, K_{iw}$ can be adjusted according to Equation (12):

$$\begin{cases} K_{pw} = \frac{J\beta}{1.5 p_n \varphi_f} \\ K_{iw} = \beta K_{pw} \end{cases} \tag{12}$$

After conducting experimental debugging and replacing the motor parameters, it was found that using the simplest deviation PI adjustment can achieve the desired outcome, where $\beta$ takes a value of 500, $\xi_a$ takes a value of 0, $K_{pw} = 0.004$, and $K_{iw} = 2$.

After the Clarke and Park transformations, the current equation of PMSM can be expressed as follows:

$$\begin{cases} \frac{di_d}{dt} = \frac{1}{L_d}(-Ri_d + L_q \omega_e i_q + u_d) \\ \frac{di_q}{dt} = -\frac{1}{L_d}[Ri_d + \omega_e(L_d i_d + \varphi_f) - u_q] \end{cases} \tag{13}$$

However, it is clear from Equation (13) that the cross-coupled electromotive force generated by the stator current $i_d, i_q$ in the d- and q-axes, respectively, must be removed:

$$\begin{cases} u_{d0} = u_d + \omega_e L_q i_q = Ri_d + L_d \frac{di_d}{dt} \\ u_{q0} = u_q - \omega_e(L_d i_d + \varphi_f) = Ri_q + L_q \frac{di_q}{dt} \end{cases} \tag{14}$$

where $u_{d0}, u_{q0}$ represents the voltage applied to the magnetic flux of the PMSM rotor after decoupling the current.

We then performed the Raschel transformation of Equation (14):

$$I(s) = G(s)U(s) \tag{15}$$

Among them: $I(s) = \begin{bmatrix} i_d(s) \\ i_q(s) \end{bmatrix}, G(s) = \begin{bmatrix} R + sL_d & 0 \\ 0 & R + sL_q \end{bmatrix}^{-1}, U(s) = \begin{bmatrix} u_{d0}(s) \\ u_{q0}(s) \end{bmatrix}.$

In this paper, a conventional PI regulator is used in conjunction with a feed-forward decoupling control strategy to obtain the *d-q* axis voltage:

$$\begin{cases} u_d^* = (K_{pd} + \frac{K_{id}}{s})(i_d^* - i_d) - \omega_e L_q i_q \\ u_q^* = (K_{pq} + \frac{K_{iq}}{s})(i_q^* - i_q) - \omega_e(L_d i_d + \varphi_f) \end{cases} \tag{16}$$

where $K_{pd}, K_{pq}$ and $K_{id}, K_{iq}$ are the PI controller's proportional and integral gains, respectively.

Due to the advantages of a single parameter and convenient adjustment, the IMC is chosen to replace the feedforward decoupling control strategy for parameter tuning design.

Figure 2a shows the basic structure of the IMC, where $\hat{H}(s)$ is the internal mold, $H(s)$ is the controlled object, and $C(s)$ is the internal mold controller. An appropriate equivalent

transformation of Figure 2a leads to the schematic block diagram shown in Figure 2b with an equivalent controller:

$$F(s) = \left[I - C(s)\hat{H}(s)\right]^{-1} C(s) \tag{17}$$

where $I$ is the unit matrix.

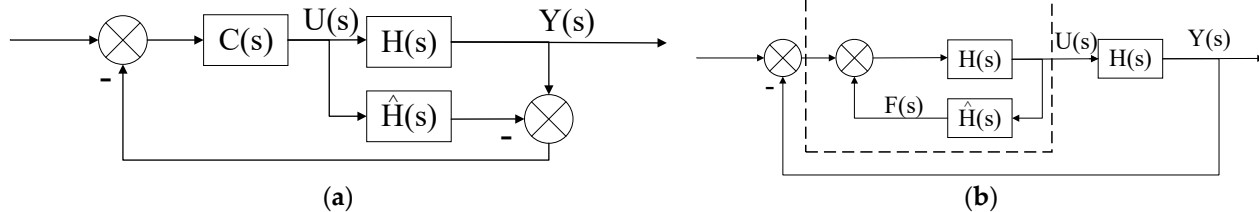

**Figure 2.** Diagram of internal mold control structure. (**a**) Block diagram of IMC; (**b**) equivalent block diagram of IMC.

When the internal modeling is accurate, $\hat{H}(s) = H(s)$, there is no feedback link in the system, when the system transfer function is:

$$G_c = C(s)H(s) \tag{18}$$

So, the system is stable when and only when $H(s)$ and $C(s)$ are stable.

The current loop of the control system can be approximated as a first-order system by considering the significant difference in time constants, particularly those related to electromagnetic and mechanical properties. According to $\hat{H}(s) = H(s)$, the definitions are as follows:

$$C(s) = \hat{H}^{-1}(s)L(s) = H^{-1}(s)L(s) \tag{19}$$

where $L(s) = \frac{aI}{s+a}$; $a$ is the design parameter, and the formula is: $\frac{2\pi R}{L}$, where $L$ is the inductance and $R$ is the resistance.

Substituting Equation (19) into Equation (17) yields the internal mode controller:

$$F(s) = a\begin{bmatrix} L_d + \frac{R}{S} & 0 \\ 0 & L_q + \frac{R}{S} \end{bmatrix} \tag{20}$$

Substituting Equation (20) into Equation (18), we obtain:

$$G_c(s) = \frac{a}{s+a}I \tag{21}$$

A comparison of Equation (21) and Equation (16) is then obtained:

$$\begin{cases} K_{pd} = aL_d \\ K_{id} = aR \\ K_{pq} = aL_q \\ K_{iq} = aR \end{cases} \tag{22}$$

The final values were obtained by substituting the motor parameters and adjusting them during the experiment, respectively: $K_{pd} = K_{pq} = 120.54$; $K_{id} = K_{iq} = 70{,}440$.

The simulation model of the continuous PI regulator is built according to Equations (16) and (22). At this point, the parameterization of the PI regulator is completed.

In the process of experimental simulation parameter adjustment, it was found that omitting "$\omega_e L_q i_q$" and "$\omega_e(L_d i_d + \varphi_f)$" in Equation (16) can also achieve a good control effect, i.e., Equation (16) is rewritten as Equation (23):

$$\begin{cases} u_d^* = (K_{pd} + \frac{K_{id}}{s})(i_d^* - i_d) \\ u_q^* = (K_{pq} + \frac{K_{iq}}{s})(i_q^* - i_q) \end{cases} \tag{23}$$

### 3. ADRC Design and Parameterization

The SMO-based motor control method, on the one hand, invokes discontinuous switching functions in the SMO bringing about high-frequency jitter. The presence of jitter means the rotor angle estimated by the observer has a large error, and this error also affects the motor speed at all times. On the other hand, traditional PI controllers are utilized in both the speed and current loops of this control method to regulate the speed. The PI controller only roughly selects the deviation between the input and output, which leads to a significant speed deviation during motor startup, and thus highly prone to overshoot.

Therefore, in order to reduce the speed overshoot during motor starting and the high-frequency vibration caused by SMO, we further introduced ADRC to improve the motor speed ring. The three closed-loop control links are reduced by ADRC compared to SMO, and the entire control system's immunity is improved. In order to achieve high-precision speed tracking while dealing with a sudden shift in PMSM load, the ADRC's accurate estimation and compensating capabilities are also put to use.

ADRC consists of three parts: TD, ESO, and NLSEF. Figure 3 displays the system's general control block diagram. Figure 3 is the overall schematic diagram of the ADRC control system.

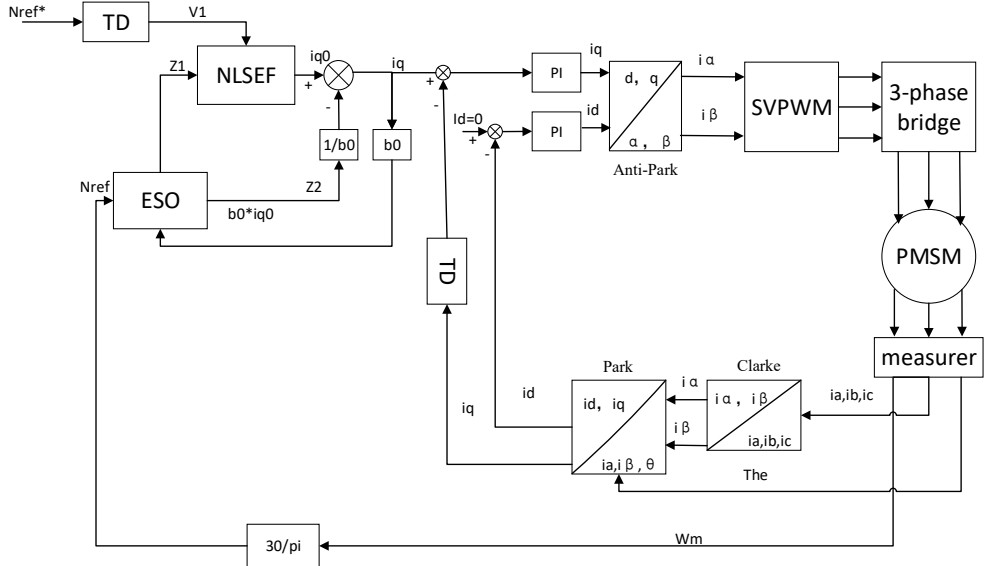

**Figure 3.** PMSM speed control system based on ADRC.

The PMSM model of Equation (2) is first expressed as a "series-integral type", which is obtained by transforming the two equations in Equation (2):

$$\frac{d^2\omega_m}{dt^2} = -\frac{dT_L}{Jdt} - \frac{Bd\omega_m}{Jdt} - \frac{K_t R_s}{JL_s}i_q - \frac{n_p K_t \varphi_f}{JL_s}\omega_m + \frac{K_t}{JL_s}u_q \qquad (24)$$

Equation (4) is converted to the following form:

$$\frac{d^2\omega_m}{dt^2} = bu_q + a(t) \qquad (25)$$

In the above equation, $b = \frac{K_t}{JL_s}$ denotes the control gain in the motor model; $a(t)$ denotes the total disturbance. Its specific form is shown in Equation (26):

$$a(t) = -\frac{dT_L}{Jdt} - \frac{Bd\omega_m}{Jdt} - \frac{K_t R_s}{JL_s}i_q - \frac{n_p K_t \varphi_f}{JL_s}\omega_m + f(t) \qquad (26)$$

where $f(t)$ is the other unknown perturbation, which is not taken into account in the current algorithm.

ADRC is the standard-type structure that realizes Equation (7) by estimating the total perturbation $a(t)$ in real time using an expansion observer and compensating it.

$$\frac{d^2\omega_m}{dt^2} = bu_q \tag{27}$$

Therefore, the estimation performance of the expansion observer is a key indicator of the control performance of the whole system. In the following, each of the three parts of the ADRC will be designed to analyze and determine the parameters to be adjusted and the direction of adjustment.

*3.1. TD Design*

As an important component of ADRC, the TD mainly enables the differential extraction of rotational speed signals for motor systems, tracks the given speed, and avoids the amplification of noise that occurs with classical differentiation. The speed overshoot issue brought on by the usage of PI controllers in the current loop is improved by the TD's arrangement of the transition process to take into consideration the "tolerance" of the motor control object. The transition mechanism set up by the PMSM speed control system for a particular $\overline{\omega}_m$ is represented by Equation (28). The following methods can be utilized with the PMSM speed control system:

$$\begin{cases} fh = fhan\left[\omega_{ref}(t) - \overline{\omega}_m, \dot{\omega}_{ref}(t), r_0, h\right] \\ \omega_{ref}(t+h) = \omega_{ref}(t) + h \cdot \dot{\omega}_{ref}(t) \\ \dot{\omega}_{ref}(t+h) = \dot{\omega}_{ref}(t) + h \cdot fh \end{cases} \tag{28}$$

where $\overline{\omega}_m, \omega_{ref}, \dot{\omega}_{ref}$ stands for the specified angular velocity, the transition process between that value's differential value and the real angular velocity, $r_0$ represents the speed parameter, and $h$ represents the filtering parameter. It typically takes the ADRC calculation period of two to obtain $h = 10 \times 10^{-4}$. Equation (8) is the form of the nonlinear TD using the most rapid synthesizing function $fhan()$. Its discrete expression is given below:

$$f = fhan(x_1, x_2, r, h) = \begin{cases} d = rh^2, a_0 = hx_2, y = x_1 + a_0 \\ a_1 = \sqrt{d(d+8|y|)} \\ a_2 = a_0 + sign(y)(a_1 - d)/2 \\ s_y = (sign(y+d) - sign(y-d))/2 \\ a = (a_0 + y - a_2)s_y + a_2 \\ s_a = (sign(a+d) - sign(a-d))/2 \\ f = -r(\frac{a}{d} - sign(a))s_a - rsign(a) \end{cases} \tag{29}$$

where $x_1, x_2$ represents the inputs, $f$ is the output, and $r$ is the limit value of the output $f$. After system debugging, when $r_0$ is set to 100,000, the system has a faster response speed and improved tracking performance.

Since the TD itself also has a filtering function to reject wild values, this paper introduces a TD in the real feedback loop of $i_q$ to perform smoothing of the $i_q$ signal and improve the control accuracy of the current.

*3.2. ESO Design*

In cases where both internal and external disturbances manifest in the physical motor system, ESO enables the simultaneous observation and compensation of these disturbances as a unified "total disturbance" to mitigate vibration. Additionally, ESO facilitates the expansion of speed information into a novel state variable for precise speed-tracking observation, thereby establishing conducive conditions for ensuring overall system stability.

The angular velocity $\hat{\omega}_m$ and its differential $\hat{\dot{\omega}}_m$ as well as the total perturbation are determined using the nonlinear ESO based on the second-order model in Equations (24)–(27) and the definition of the total perturbation $a(t)$, where $fal(e, \alpha, \delta)$ is a nonlinear function, as shown by Equation (30).

$$fal(e, \alpha, \delta) = \begin{cases} \frac{e}{\delta^{1-\alpha}}, |e| \leq \delta \\ sign(e)|e|^{\alpha}, |e| > \delta \end{cases} \tag{30}$$

where $fal(e, \alpha, \delta)$ is an idempotent function; $\alpha$ is the power; and $\delta$ is a linear interval. The nonlinear ESO is realized in the form of the following Equation (31).

$$\begin{cases} e(t) = \hat{\omega}_m(t) - \omega_m(t) \\ fe(t) = fal(e(t), \alpha_1, \delta) \\ fe_1(t) = fal(e(t), \alpha_2, \delta) \\ \hat{\omega}_m(t+h) = \hat{\omega}_m(t) + h(\hat{\dot{\omega}}_m(t) - \beta_1 e(t)) \\ \hat{\dot{\omega}}_m(t+h) = \hat{\dot{\omega}}_m(t) + h(z(t) - \beta_2 fe(t)) + h b_0 u_q(t) \\ z(t+h) = z(t) + h(-\beta_3 fe_1(t)) \end{cases} \tag{31}$$

where $\omega_m$ is the feedback's actual angular velocity, $\hat{\omega}_m, \hat{\dot{\omega}}_m$ is the predicted value for the feedback's angular velocity and its differentiation, $z$ is the overall perturbation value estimated in real time, $\beta_1, \beta_2, \beta_3$ is the ESO's correction gain, $b_0$ is the compensation factor, and $u_q$ is the voltage on the rotor of a motor. The ESO's settings that need to be adjusted are $\alpha, \delta, \beta_1, \beta_2, \beta_3$. The effects of this parameter on the system are analyzed one by one below.

### 3.2.1. α and β Parameterization

The following transformation is applied to the idempotent function in Equation (30):

$$\begin{cases} \lambda(e) = \frac{fal(e, \alpha, \delta)}{e} \\ fe(t) = \frac{fal(e, \alpha_1, \delta)}{e} e = \lambda_1(e)e \\ fe_1(t) = \frac{fal(e(t), \alpha_2, \delta)}{e} e = \lambda_2(e)e \end{cases} \tag{32}$$

From the above equation, it can be seen that parameters $\alpha$ and $\delta$ are both important and interrelated to the system performance, respectively.

When $\alpha$ is a constant, the linear interval—that is, the range in which the output of function $\lambda(e)$ is constant—gets smaller as $\delta$ decreases. The smaller $\delta$ is, the larger the output value of function $\lambda(e)$ is at small errors, but when $\delta$ is too small, the gain is too large, which is likely to cause system instability. In contrast, as $\delta$ increases, it fails to highlight the advantages of nonlinearity. Therefore, in practice, it is necessary to determine a suitable linear interval for the system, which is generally taken as $0.01 < \delta < 0.1$, here $\delta = 0.01$.

When $\delta$ is a fixed value, the power $\alpha$ value is smaller, the constant value of the function $\lambda(e)$ is larger, and the nonlinear "large error, small gain; small error, large gain" characteristics are more obvious. The control volume could oscillate at high frequencies if $\alpha$ is set too low, and if it is set too high, the fast error decay and strong disturbance immunity will not be utilized. Statistically, $\alpha_1 = 0.95$ and $\alpha_2 = 0.5$.

### 3.2.2. ESO Gain Adjustment

In practical motor systems, ESO is capable of simultaneously observing and compensating for internal and external disturbances, avoiding the occurrence of vibration phenomena. Additionally, ESO converts the speed information into new state variables to accurately track the motor speed, thereby ensuring system stability.

There are several feasible methods for the tuning of parameter $\beta_1, \beta_2, \beta_3$ in a nonlinear ESO. One of the most classical methods is to replace the three parameters $\beta_1, \beta_2, \beta_3$ with a

single-parameter-observer bandwidth $\omega_0$, and the tuning of the ESO gain can be realized by adjusting $\omega_0$ directly. The specific formula of the system is shown in Equation (33):

$$\begin{cases} \beta_1 = 6\omega_0 \\ \beta_2 = \frac{3\omega_0}{2} \\ \beta_3 = \frac{3\omega_0^2}{50} \end{cases} \tag{33}$$

The larger $\omega_0$, the higher the estimation accuracy of the observer, the stronger the immunity to interference, and the faster the transient response. However, too large $\omega_0$ will also bring high-frequency noise or ESO oscillation. The comprehensive performance of this system is taken as $\omega_0 = 1000$, combined with Equation (33), and after parameterization, is taken as $\beta_1 = 55,000$, $\beta_2 = 36,000$, $\beta_3 = 53,000$.

### 3.3. NLSEF Design

In the NLSEF proposed in this paper, the system's state error is defined as the difference between the estimated outputs of the ESO's load perturbation and speed state and the outputs of the TD:

$$e_1 = v_1 - z_1 \tag{34}$$

where $z_1$ represents the estimated rotational speed, $z_2$ represents the estimated load disturbance, and $v_1$ represents the estimated rotational speed tracking signal.

The fractional linear PID combination is then transformed into a series controller form, which may be derived as the equivalent NLSEF form by differentiating the *Nref* tracking signal of the TD from the *Nref* observation signal in the observer:

$$\begin{cases} e_1 = v_1 - z_1 \\ iq_0 = K \cdot fal(e_1, \alpha_1, \delta) \\ iq = iq_0 - \frac{1}{b \cdot z_2} \end{cases} \tag{35}$$

where $i_{q0}$ is the preset current value; $i_q$ is the current value after adding the feedforward disturbance; $K$ is the gain parameter for ADRC, and $b$ is the disturbance compensation parameter. In this paper, $\alpha_1$ and $\delta$ are taken as 0.95 and 0.01, respectively, in the nonlinear controller.

Precise observation and appropriate correction of the overall system disturbance form the foundation of ADRC. The compensation parameter $b$ reflects the compensation strength of the controller. Increasing $b$ can reduce the phenomenon of system oscillation, but it may also deteriorate the control performance of the system. In this paper, we first take $b$ in Equation (25), which is calculated by the model, as the base value, and keep increasing it until finally taking $b = 5000$ as appropriate.

The larger the system control gain $K$, the shorter the system settling time. However, too large a value of $K$ may lead to system overshoot or even oscillation. After the disturbance compensation factor $b$ has been established, the control gain $K$ can then be suitably modified in accordance with how the system is operating. The control gain used in this paper is $K = 4$ following system debugging.

### 3.4. Fuzzy-ADRC Design

In practice, the system receives a small disturbance from the outside world, which will cause the motor speed to decrease and not reach the set speed value after ADRC closed-loop control and there is still a gap in the set speed. The motor's measured speed value and set value error are both tiny, which makes it challenging to employ closed-loop control to produce the necessary regulating effect. The speed error can be scaled up by introducing a proportionality factor and then passing it to the ADRC for further control. In order to achieve flexible adjustment of the proportionality coefficient, this paper introduces FC and utilizes the fuzzy rule of "small error, large gain". The obtained small error in rotational speed is amplified by the proportional coefficient and then transmitted to the

ADRC system. The AC PMSM servo system's robustness and disturbance compensation are significantly enhanced by this approach.

The FC used in this paper utilizes the ability of the adaptive method of fuzzy control to enable the controller to provide the best estimate of the desired parameters within a certain range. First, the system's fuzzy inference rules are formulated. Then, a fuzzy rule table is created, utilizing the fuzzy subset affiliation function for the velocity error, acceleration, and the fuzzy control model of each parameter. This fuzzy rule table is designed to effectively compensate for minor disturbances within the system. After defuzzification, the necessary proportional amplification factor for the speed error can be identified and paired with the regulation of the ADRC and PI controllers to determine the motor's final speed output. Figure 4 is the overall schematic diagram of the Fuzzy-ADRC control system.

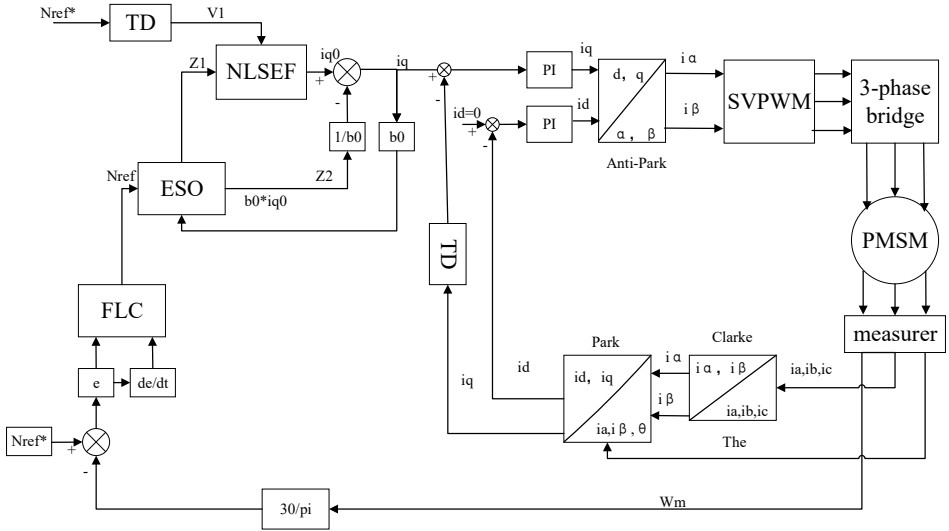

Nref*: Given rotational speed.  Nref: Measured rotational speed.

**Figure 4.** PMSM speed control system with Fuzzy-ADRC.

Where err is the output variable and r and rc are the fuzzy input variables for the controller, respectively. In their domain S, seven subsets of languages are specified as "negative large (n3)", "minus medium (n2)", "negative small (n1)", "zero (O)", "positive small (p1)", "plus medium (p2)", and "positive large (p3)". The fundamental domains of the input quantities r and rc are $[-1000, 1000]$ and $[-100, 100]$, respectively, and the fundamental domain of the output quantity err is $[-100, 100]$. The affiliation function of each variable has seven grades corresponding to the fuzzy linguistic variables, and the affiliation function curves are in the form of a combination of affiliation functions with the Gaussian normal distribution type (gaussmf) on both sides and the triangular type (trimf) in the middle.

The defuzzification processing methodology uses the average weighting method, and the fuzzy inference is of the Mamdani type. Table 2 displays a list of fuzzy control rules for determining parameter rectification.

**Table 2.** Table of fuzzy control rules.

| r/rc | n3 | n2 | n1 | O | p1 | p2 | p3 |
|------|----|----|----|----|----|----|----|
| n1 | O | O | O | n1 | n2 | n2 | n3 |
| n2 | O | n1 | n1 | n1 | n2 | n3 | n3 |
| n1 | n1 | n1 | n2 | n2 | n3 | n3 | p2 |
| O | n2 | n2 | p3 | p3 | p3 | p2 | p2 |
| p1 | n2 | p3 | p3 | p2 | p2 | p1 | p1 |
| p2 | p3 | p3 | p2 | p1 | p1 | p1 | O |
| p3 | p3 | p2 | p1 | p1 | O | O | O |

After system debugging, it was found that the proportional coefficient value output is too large to cause the system oscillation caused by speed overshooting, so in this paper, r and rc were multiplied by coefficients of 0.001. The following equation is constructed to represent the motor speed of the fuzzy control output:

$$Nref = (30/pi) * 1.18\omega_m + 0.1err \tag{36}$$

The values of the final parameters of the simulation model constructed in this paper are listed in Table 3.

**Table 3.** Summary of control parameters for the simulation model.

| Parameters | Numerical Value | Parameters | Numerical Value |
|---|---|---|---|
| $\beta$ | 500 | $r_0$ | 100,000 |
| $K_{pw}$ | 0.004 | $\delta$ | 0.01 |
| $K_{iw}$ | 2 | $\alpha$ | 0.95 |
| $K_{pd}$ | 120.54 | $\beta_1$ | 55,000 |
| $K_{pq}$ | 120.54 | $\beta_2$ | 36,000 |
| $K_{id}$ | 70,440 | $\beta_3$ | 53,000 |
| $K_{iq}$ | 70,440 | $b$ | 5000 |
| $h$ | $10 \times 10^{-4}$ | $K$ | 4 |

## 4. Analysis of Simulation Findings

In this study, the MATLAB2021b/Simulink environment is used to create the motor simulation model. The simulation model of PMSM speed control based on Fuzzy-ADRC under small disturbance is shown in Figure 5. The simulation time of the system is set to 0.5 s, and the variable step size automatic solving algorithm is selected. The main modules used in the simulation model are a DC power supply, a general-purpose bridge, a PMSM motor module, a Clark coordinate transformation module, a Park coordinate transformation module, an inverse Park coordinate transformation module, an SVPWM module, a PI controller, an ADRC module, FC, and so on.

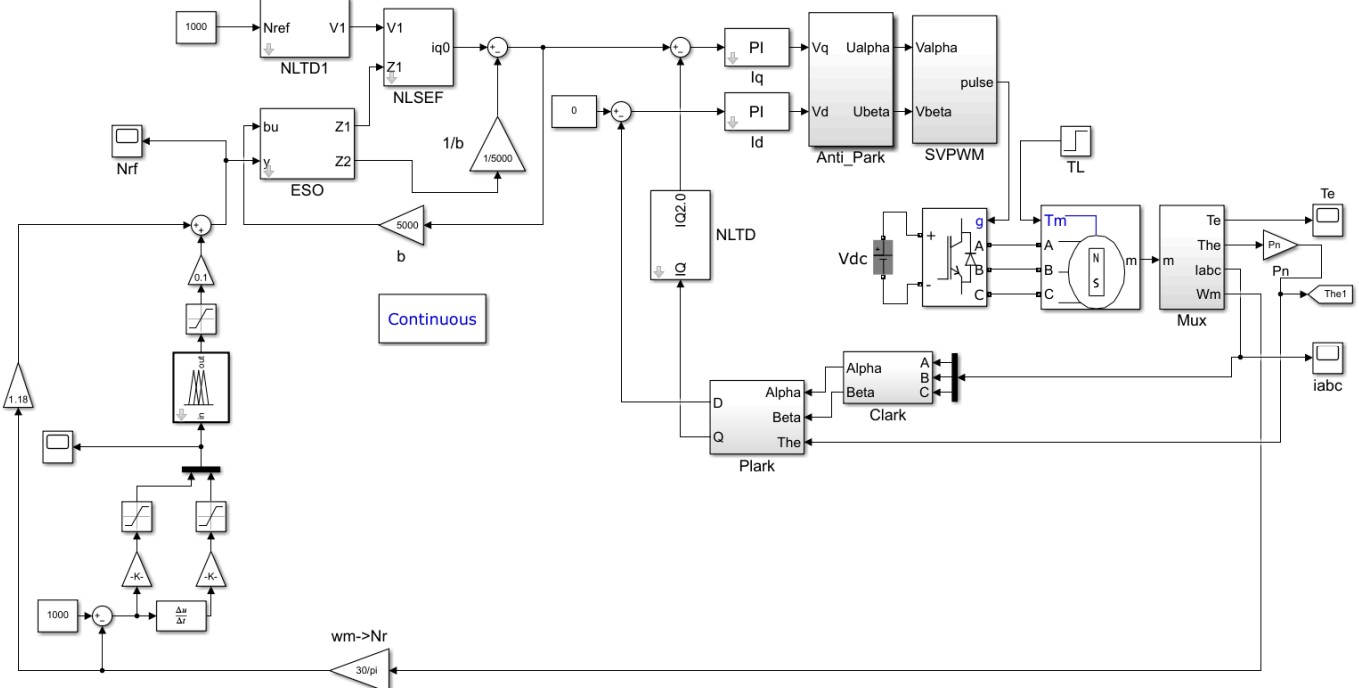

**Figure 5.** Simulation model of PMSM speed control based on Fuzzy-ADRC under small perturbation.

An actual PMSM is chosen for simulation in this research to ensure that the algorithm is functional. The motor model is 60ST-M00630, and Table 4 lists the details of the motor.

**Table 4.** Main parameters of PMSM.

| Parameters | Numerical Value |
|---|---|
| Rated voltage U/V | 220 |
| Rated power P/W | 200 |
| Rated speed n/(r/min) | 3000 |
| Rated torque Te/(N·M) | 10 |
| Rated current I/A | 1.5 |
| Rotor inertia J/(kg·m$^2$) | $0.17 \times 10^{-4}$ |
| Permanent magnet flux $\varphi_f/\omega_b$ | 0.3477 |
| Stator resistance (between lines) Rs/$\Omega$ | 11.6 |
| Stator inductance (between lines) Ls/H | 0.022 |
| Switching frequency $f_s$/kHz | $1 \times 10^4$ |
| Polar logarithm P | 4 |

### 4.1. Analysis of Simulation Findings

In this section, the motor speed waveforms of three controllers, SMO, ADRC, and Fuzzy-ADRC, are compared and analyzed when the motor speed is set to 1000 revolutions per minute, as shown in Figure 6.

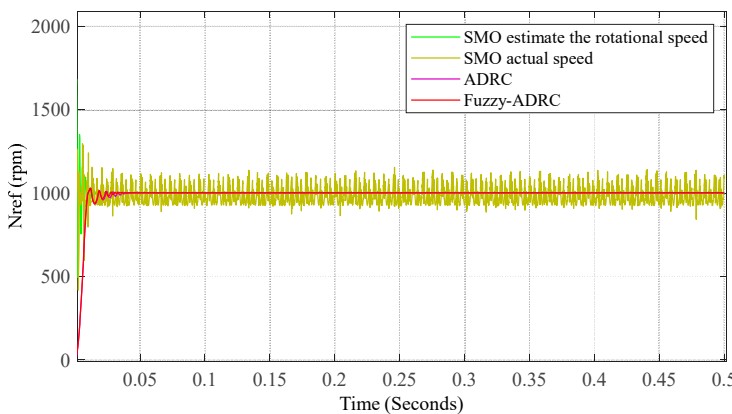

**Figure 6.** The corresponding motor speed curves of the three controllers when the set speed is 1000 rpm.

From the speed curves, it can be seen that all three controllers have different degrees of overshooting oscillations for a period of time after the motor starts. The SMO speed-exceeded percentage was measured to be 70% with a regulation time of 0.012 s, the ADRC speed-exceeded percentage was 4.1% with a regulation time of 0.026 s, and the Fuzzy-ADRC speed-exceeded percentage was 2.9% with a regulation time of 0.017 s. Among them, the SMO has a large speed-exceeded percentage but a rapid dynamic response. The ADRC has a small speed-exceeded percentage but a long dynamic response time. The Fuzzy-ADRC has the smallest speed-exceeded percentage and has a shorter dynamic response time than the ADRC. Therefore, by combining the two indicators of speed-exceeded percentage and regulation time, the Fuzzy-ADRC has the lowest speed-exceeded percentage and better dynamic performance than the remaining two controllers.

### 4.2. Introduction of Small Load Experimental Simulation Analysis

During the process of adjusting the motor speed to 1000 rpm, a load torque of 5 Nm was increased over the duration of 0.1 s. We performed a comparative analysis of the motor speed waveforms under small load disturbances for the three controllers, SMO, ADRC, and Fuzzy-ADRC, as shown in Figure 7.

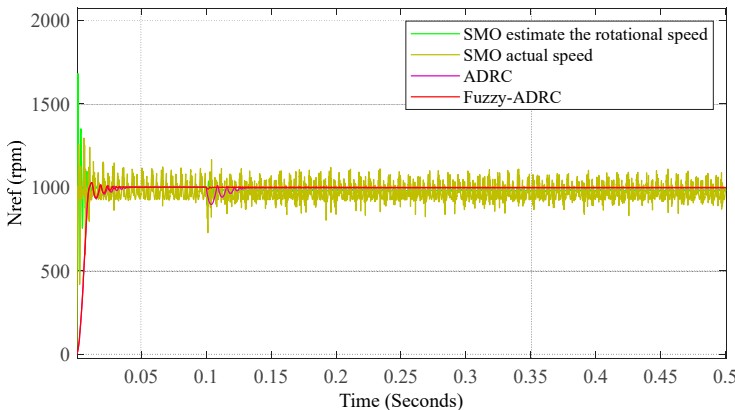

**Figure 7.** Set the initial speed to 1000 rpm and add 5 loads at 0.1 s.

Based on Figure 7, it is clear that different controllers experience a temporary decrease in speed when the load torque is increased. Nevertheless, with internal adjustments, the controllers eventually achieve a stable speed that aligns with the desired set value. The SMO was measured to maintain the speed at 980 rpm, the ADRC at 990 rpm, and the Fuzzy-ADRC at 998 rpm. This suggests that the SMO has a quick dynamic response yet a high landing speed. The speed of landing of ADRC is only less than that of the SMO. The Fuzzy-ADRC has the least speed landings and is almost unaffected by load disturbances. To summarize, the Fuzzy-ADRC has the capacity to efficiently compensate for and rectify minor disturbances. It accurately observes external small perturbations and provides suitable compensation, surpassing the compensation abilities of ADRC.

### 4.3. Simulation Analysis under Variable Speed Conditions

At the initial moment, the motor speed was set to 800 rpm. At 0.15 s, the set speed changed to 600 rpm, and at 0.3 s, the set speed changed to 1000 rpm. We compared and analyzed the motor speed waveforms of three controllers, SMO, ADRC, and Fuzzy-ADRC, as shown in Figure 8.

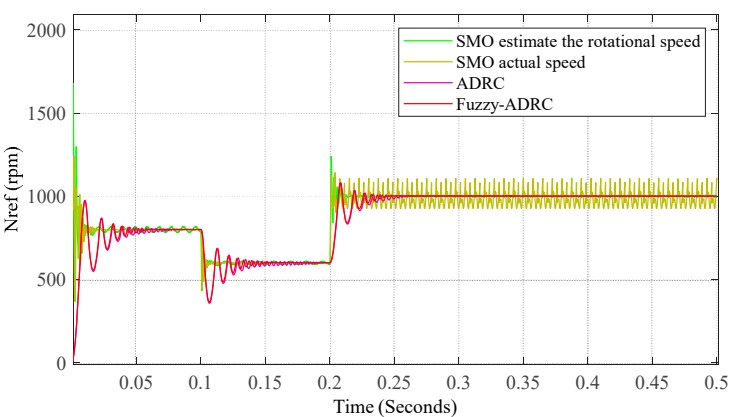

**Figure 8.** Speed profiles corresponding to the three controllers during motor acceleration and deceleration.

From the above motor speed curves, it can be seen that by changing the speed setpoints in different time periods, the PMSMs under the three controllers can eventually stabilize at the set speed values after a short period of oscillation. Based on the findings shown in Figure 8, it is evident that the SMO exhibits faster regulation during both motor acceleration and deceleration. However, the continuous oscillations in speed can be directly linked to the inherent deficiencies of the sliding-mode control algorithm. Both Fuzzy-ADRC and ADRC have smoother speeds after a short period of oscillation, but the regulation time of ADRC is quite long. Therefore, ADRC with FC provides smoother speed control than SMO.

### 4.4. Simulation Analysis with the Introduction of Variable Loads

In order to investigate the impact of load variation on the performance of the Fuzzy-ADRC system, we examined the load variation during motor operation. To examine the motor speed tracking, the system set the initial speed to 1000 revolutions per minute, applied a load torque of 15 Nm at 0.15 s, and then removed the load torque at 0.3 s. Figure 9 illustrates the corresponding variations in motor speed and winding current as the load changes.

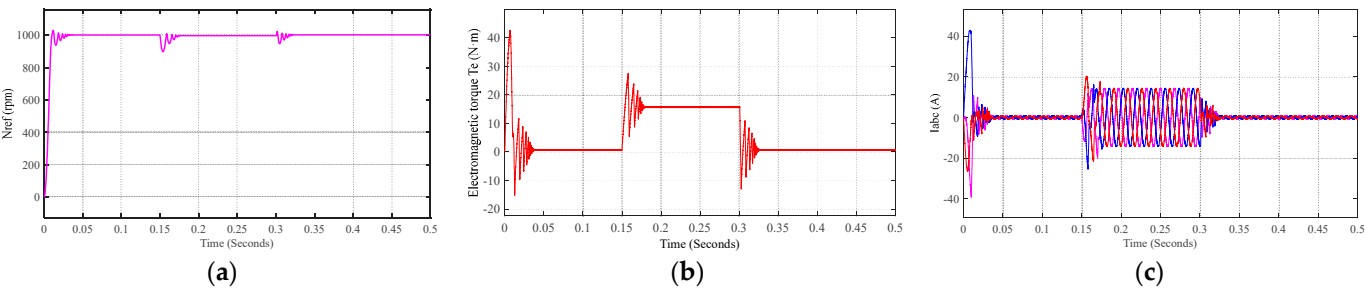

**Figure 9.** State tracking of Fuzzy-ADRC control during variable motor load. Notes: (**a**) Velocity tracking; (**b**) variable load; (**c**) the three-phase current waveform of motor windings.

According to the observations shown in Figure 9a, under Fuzzy-ADRC operation control, the speed of the motor remains relatively stable when the load torque is introduced. After measurement, the Fuzzy-ADRC initially stabilizes the motor speed at 1003 rpm; the speed becomes 996 rpm after adding the load torque, with a difference of 5 rpm; and the speed is restored to 1003 rpm after removing the load torque at 0.3 s. This results in a Fuzzy-ADRC with better load-carrying capability.

This section introduces the use of variable loads to achieve load superposition and further validate the accurate compensation of Fuzzy-ADRC for minor disturbances in PMSM. To observe the motor speed tracking, the system applies a load torque of 20 Nm at 0.1 s, 15 Nm at 0.2 s, and 10 Nm at 0.3 s. Figure 10 illustrates the corresponding variations in motor speed and winding current during the load superposition process.

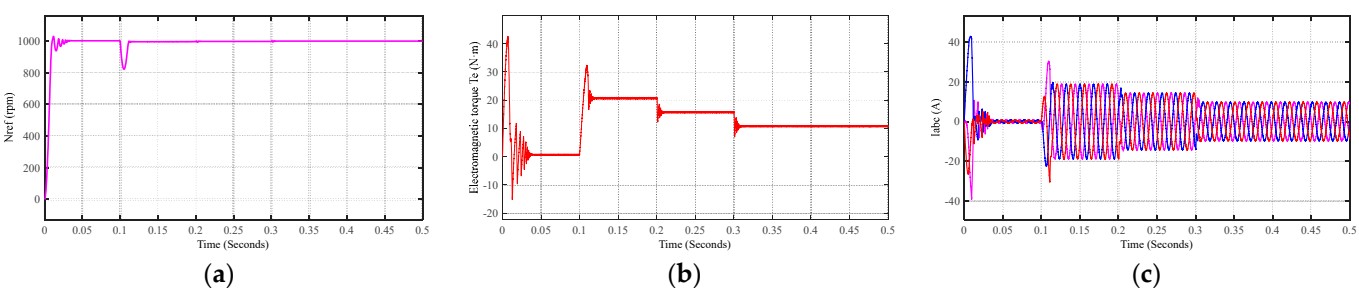

**Figure 10.** Effect curve of Fuzzy-ADRC control under motor load superposition. Note: (**a**) Velocity tracking; (**b**) load stacking; (**c**) the three-phase current waveform of motor windings.

It is clear from Figure 10a that changing load values for the Fuzzy-ADRC controller will result in varying oscillation and speed landing intensities; the larger the load, the more intense the oscillation, and the speed landing also rises. The motor's starting speed was determined to be 1003 rpm. By applying 20, 15, and 10 Nm load torques, respectively, at intervals of 0.1, 0.2, and 0.3 s, speeds of 993, 996, and 998 rpm were attained. The motor's rated load was tested and found to be 20 Nm. This experiment demonstrates the efficacy of the fuzzy self-tuning-based Fuzzy-ADRC technique for ensuring high-precision speed tracking of the PMSM servo system.

### 4.5. Simulation Analysis of Internal Parameter Variations in Motors

To further explore the influence of internal parameter changes on the control effectiveness using the Fuzzy-ADRC strategy, this section compares and analyzes the speed curves. Specifically, we vary two specific internal parameters of the motor, namely, the magnetic flux coefficient and the stator inductance, for evaluation. Figure 11 illustrates the effect on motor speed as the magnetic flux coefficient or stator inductance value changes.

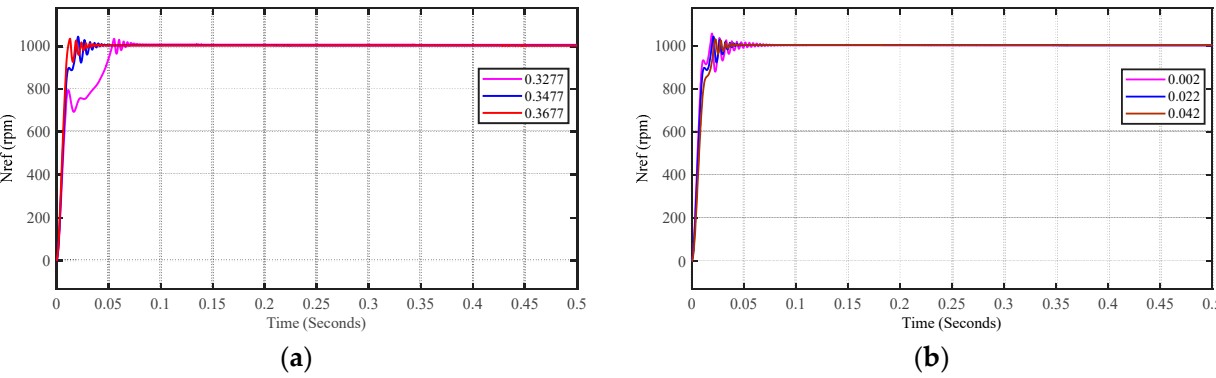

**Figure 11.** The variations in internal parameters correspond to changes in the speed curves. Note: (**a**) Varying the magnetic flux coefficient; (**b**) changing the stator inductance value.

The legend in Figure 11 represents the parameters of the corresponding motor. The measurements indicate that when the magnetic flux coefficient is set at 0.3277, 0.3477, and 0.3677, the system exhibits dynamic response times of 0.051 s, 0.019 s, and 0.017 s, respectively. Similarly, for stator inductance coefficient values of 0.002, 0.022, and 0.042, the dynamic response times are 0.031 s, 0.017 s, and 0.020 s. These measurements suggest that within a specific range of changes, increasing the magnetic flux coefficient or stator inductance can enhance the stability of the system, resulting in smoother operation and indirectly accelerating the dynamic response. Conversely, reducing these parameters may lead to a slower dynamic response of the system.

## 5. Conclusions and Outlook

In this study, we propose a Fuzzy-ADRC-based speed controller for a PMSM designed to handle small disturbances. To evaluate its performance, we develop and compare three motor control simulation models: SMO, ADRC, and Fuzzy-ADRC. The validation results show that Fuzzy-ADRC outperforms SMO and ADRC in terms of settling time and disturbance compensation. Fuzzy-ADRC achieves a smaller overshoot than SMO, reducing it from 70% to 2.9% when targeting the same motor speed. It also exhibits a faster dynamic response than ADRC, improving the settling time from 0.026 s to 0.017 s. During motor acceleration and deceleration, Fuzzy-ADRC has an average overshoot of 14%, while SMO has an average overshoot of 23%, indicating Fuzzy-ADRC's superior performance in startup and braking scenarios. Additionally, Fuzzy-ADRC provides smoother speed control and demonstrates minimal speed drop and high compensation accuracy when a small load torque is introduced, remaining largely unaffected by load disturbances. By appropriately amplifying the speed error in the output coefficients of the FC in the presence of slight variations in the motor's internal parameters, ADRC's compensation ability can be enhanced, thus improving the system's robustness against internal disturbances. Consequently, the Fuzzy-ADRC speed control strategy exhibits enhanced robustness, anti-interference capability, and reduced sensitivity to system characteristics and external disturbances. However, it is important to note that this study focuses solely on simulating small disturbances. Future research will further investigate the influence of iron core saturation and motor parameter variations on system control performance in real-world motors, using methods such as parameter identification. Additionally, expanding the

range of considered disturbances will contribute to improving the practicality of this control algorithm.

**Author Contributions:** Conceptualization, Q.Z.; methodology, Q.Z.; software, C.Z.; validation, C.Z.; formal analysis, Q.Z.; investigation, C.Z.; resources, C.Z.; data curation, C.Z.; writing—original draft preparation, C.Z.; writing—review and editing, Q.Z.; visualization, Q.Z.; supervision, Q.Z.; project administration, Q.Z.; funding acquisition, Q.Z. All authors have read and agreed to the published version of the manuscript.

**Funding:** This research was funded by the key research and development projects in Guangxi, grant number 21076005.

**Institutional Review Board Statement:** Not applicable.

**Informed Consent Statement:** Not applicable.

**Data Availability Statement:** Not applicable.

**Acknowledgments:** The authors would like to thank the Intelligent Control Base Laboratory of Guilin University of Electronic Science and Technology for technical support.

**Conflicts of Interest:** The authors declare no conflict of interest.

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
