# Peer review of "Speed Control of PMSM Based on Fuzzy Active Disturbance Rejection Control under Small Disturbances"

_applsci, doi:10.3390/app131910775_

Round 1

Reviewer 1 Report

The topic of this paper is very interesting, however there are some shortcomings which should be improved by considering the following points:

1.     There are some grammatical mistakes, misspellings and unclear expressions in text.

2.     I suggest that a table (or list) should be added to show all abbreviations and parameters. The aim is to read conveniently.

3.     The “Introduction" should be greatly improved. I suggest that a comparison of your work with references should be given to clearly show your aim, innovation, and contribution.

4.     Please carefully check all parameters of the full text. They should be consistent in figures, equations, and tables.

5.     Some references should be labelled when some others’ works are cited, especially in the introduction.

6.     Line 349: Title table needs to be revised. "Table 2. This is a table. Tables should be placed in the main text close to the first time they are cited."

7.     Line 352: The coefficient 0.001 is repeated twice in line 352.

8.     Line 332: the text in this line must be revised.

9.     The figure's font should be adjusted to enhance clarity. 

Author Response

Please review the latest attachment.

Reviewer 2 Report

The paper presents a Fuzzy Active Disturbance Rejection Control approach for speed control of a permanent magnet synchronous motor. The paper is not well prepared, and there are some comments that the author should consider to improve the paper's quality.

1. Section SMO-based PMSM simulation modeling

SMO is the Sliding mode control defined in line 30? However, the content of this section is about PI controller instead of sliding mode control. An equivalent controller is mentioned in equation (17). However, I don't think it is a SMC-based one. The authors should carefully revise this section.

2. Line 285, 286: omega_0 = 1000, beta_1 = 55000, beta_2 = 36000. These values are conflict with equation (33). I doubt the accuracy of the study.

3. Line 336:  "... err is the output variable and r and rc ..." What does it mean? In line 352, the author said that r, rc were multiplied by 0.001 and 0.001. So the range of r, rc are [-1000 1000] and [-100 100] are too big. Once again, I doubt the accuracy of the study.

4. Three sub-figures in Figures 5, 6 should be merged to show the proposed control approach's effectiveness clearly. Besides, It seems that the estimated speed of SMO is not good. The question is the need for the SMO, or the model is not accurately designed?

5. Figure 5: The authors do not zoom out the Figure. I can see the exact value of the speed. However, when the system reaches a steady state, the motor speed is still greater than 1000 rpm. Why? I don't think the controllers are bad like that in simulation.

6. Subsections 3.3 and 3.4: They are still simulation results, right? The authors must replace the word "experiments". Surprisingly, with the existence of the load the system performs a higher accuracy than the no-load scenario in the steady state in which the motor speed is approximately 1000 rpm.

7. Too many abbreviations, some of which are defined many times, for example, GA, BPNN, ESO, etc. The FC is used as an abbreviation of Fuzzy Controller (line 85) and Feedforward Controller (line 86).

8. Minor issues and typos

- Line 125: "feedback current iq_ref and the supplied current iq". This statement is incorrect.

- i'q is not defined (equation 7)

- Typos in the equation (35)

- Many other typos, the authors must carefully revised.

Author Response

Please review the latest attachment.

Reviewer 3 Report

This article described the speed control of the PSMS utilizing Fuzzy-ADRC technique to improve the transient response against minor disturbances. Overall, the article is well structured and written with some comments as below:

1. The current similarity index is 24% which required further improvement. Suggest to reduce it to below 20%.

2. This article involved self citation of own conference paper up to 3% similarity. Suggest to improve and reduce it.  (Simulation of non-inductive vector control of permanent magnet synchronous motor based on sliding mode observer)

3. In this article, the disturbances here are referring to those mechanical disturbance. Any idea if other type of disturbances are involved such as electromagnetic disturbance, transient surges disturbance, etc. 

4. Please ensure the same figure caption be used in the text (Figure vs Fig.).

5. Please check carefully, some abbreviation that uses italic form should be standard throughout the text. 

6. Typo error in Table 2 (line 349)

7.  In Figure 6(c), the Fuzzy-ADRC exhibits longer settling time as compared to SMO and ADRC method. why? 

Author Response

Please review the latest attachment.

Reviewer 4 Report

1) When comparing the Fuzzy-ADRC with other control methods, consider providing quantitative metrics or performance indicators. This will strengthen the argument for the superiority of Fuzzy-ADRC and make the results more convincing.

2)  While the introduction briefly mentions the addition of a Fuzzy Controller (FC) to the rotational speed feedback loop, it would be helpful to provide a more detailed explanation of how FC integrates with ADRC and how it contributes to improved performance.

3) Discuss the practical aspects and challenges of implementing the proposed Fuzzy-ADRC system in real-world PMSM applications. Address potential limitations and considerations for deployment.

4) The paper mentions that it is a simulation test for small external perturbations. Consider discussing potential applications beyond simulations and how the algorithm may perform under different operating conditions or with larger uncertainties.

5) Explore a wider range of testing scenarios and disturbances to assess the algorithm's robustness comprehensively. Consider including scenarios with larger perturbations or complex load profiles.

Author Response

Please review the latest attachment.

Reviewer 5 Report

This manuscript designs a Fuzzy-ADRC method to control the speed of PMSM. Some concerns are as follows:

1. The authors should double-check the abbreviation of some professional phrases, e.g., avoid duplicating them, like genetic algorithm (GA); check whether 'FC' stands for Fuzzy Controller or Feedforward Control.

2. Are there any previous references touching the subject of tiny disturbance control? Why do you choose Fuzzy-ADRC as the method?

3. The Section No. needs to be reconfirmed.

4. Please check the table title. of Table 1 and the font size in Table 2.

5. Please edit the figures in terms of (a) the figure size; (b) same simulation time duration; (c) comparison of ADRC and Fuzzy-ADRC in one figure.

6. The authors should highlight the innovation of this manuscript and make it clearer both in Abstract and Introduction.

English can be improved to avoid mistakes.

Author Response

Please review the latest attachment.

Reviewer 6 Report

The article presents interesting content, but still needs some improvements. The following are my comments and observations:

1. The introduction should include a brief explanation of the chattering phenomenon.

2. Isn't the authors' decision to omit the phenomenon of saturation of machine cores an overly radical simplification? Actual ferromagnetic core machines exhibit this behavior. Is it possible for the authors to present simulation results that include approximate saturation in the motor model?

3 The authors claim that their proposed control system is robust to parameter changes. Can you please provide an analysis of the effect of changes in the parameters of the electric machine on the quality of the control?

4. Equation 4 is not fully understood. For example, the derivative of the change in inductance with respect to time appears in it, even though the authors previously assumed inductance as a constant. Please clarify this issue.

5. In line 134, instead of "magnetic chain" shouldn't it be "flux linkage"?

6. In the simulation model in Figure 4, was the motor model modeled as "sinusoidal back EMF" or as "trapezoidal back EMF"?

7. The article presents only speed variation waveforms. It is also worth adding sample waveforms of the current flowing through the motor windings in the operating cases considered by the authors.

Please take note of these comments and include the appropriate corrections in the article.

Author Response

Please review the latest attachment.

Reviewer 7 Report

The paper is concerned about simulation studies focused on speed control of a permanent magnet synchronous motor using an innovative approach, which combines active disturbance rejection algorithm with fuzzy logic controller. The subject of the paper is certainly novel and attractive for practitioners. Thus the paper might be accepted for publication, yet in the present form it requires some improvements (a more detailed discussion of some formulas and English check by a native speaker)

Eq. 1 - notation using dots for derivatives looks bad here, in particular for the currents in the refernce frame. Consider writing fractions like di/dt here

line 125 instead of "disparity" maybe write "difference"?

line 134 - do you mean magnetic flux? Formula (5) contains "theta" which is not properly explained after the first use

Eq. 9 needs some extra explanation, Why the argument of rotational inertia in the denominator is offset by beta after application of Laplace transform?

How do you determine Ld and Lq experimentally? These quantities are needed for computations e.g. in (14), (15).

line 221, rephrase this, maybe "f(t) is another perturbation present, not taken into acccount in the present algorithm"?

lines 332-333 are probably part of a rebuttal letter and not part of the text. 

line 349 this is certainly not the caption for the Table 1.

Please consult the language of your manuscript with a native speaker. It is generally ok, but I feel it might be improved a little. 

Author Response

Please review the latest attachment.

Round 2

Reviewer 1 Report

I am delighted to accept this paper in its current form.

The quality of the English language is satisfactory and meets the required standards

Reviewer 2 Report

I agree with the author's explanation. The revised manuscript is well-improved.

I recommend the paper be published after some minor modifications as:

- Figures 1, 3, and 4: All variables must be shown with subscripts. For example: ia, ib, etc.

- Figure 2: Use equation or MathType to show the correct H^(s) symbol.

- Figures 6, 7, 8, 9, and 10: The color of the SMO actual speed signal is difficult to observe. Nref(rpm) => Nref (rpm) (add space), Time(Seconds) =>Time (Seconds). The text size of the Figures should be equivalent to the main text size

- The unit N-m => Nm

Reviewer 4 Report

Paper can be accepted in present form.

Reviewer 5 Report

The authors have revised the manuscript according to the reviewers' suggestions. Some formatting still needs to be adjusted for Applied Sciences.

English has been improved.

Reviewer 6 Report

After reviewing the revised version of the article and the authors' responses, I conclude that the corrections made are insufficient at this stage to accept the paper for publication.

The authors still have not performed any experiments or analyses on the impact of the PMSM core saturation phenomenon on the quality of the control system. I understand that this aspect is often omitted from initial simulation calculations, but the lack of measurements and studies in this area remains a significant gap in the presented work. Therefore, it is necessary to carry out additional simulation analyses before accepting publication in the journal Applied Sciences.

1) My main focus is on the need to analyse the effect of PMSM parameter variations on the performance of the control system. This procedure is relatively simple to carry out, given that the authors have their own simulation model in Matlab/Simulink. All that is required is to make changes to the finished PMSM machine block, for example in parameters such as stator resistance (Rs), Armature inductance (Ls) and Flux linkage, and run the simulation. I believe that such analyses are necessary, especially given that changes in these machine parameters can occur during operation. (I pointed out this aspect in a previous review).

2) In addition, the article contains a number of editorial errors that need to be corrected. In Table 4, I note the following inaccuracies: "Stator resistance Ls [H]". (do they mean resistance in ohms?), and "Polar logarithm P = 4". - did the authors not mean the number of pole pairs ?

3) Please check the entire article for correct technical terminology in English.

Round 3

Reviewer 6 Report

The authors' completed version of the article is slightly better. Before it goes to publication, I suggest the authors make editorial corrections to both the text and the formulas. Guidelines for authors are posted on the MDPI website.

In Table 4, it is N*M and should be Nm.

In Figure 11 as well as in the text of the article, the units of the changed parameter values are missing.
